# Retrieval of TP Concentration from UAV Multispectral Images Using IOA-ML Models in Small Inland Waterbodies

**Wentong Hu, Jie Liu, He Wang, Donghao Miao, Dongguo Shao and Wenquan Gu ***

State Key Laboratory of Water Resources and Hydropower Engineering Science, Wuhan University, Wuhan 430072, China
* Correspondence: gwquan@163.com

**Abstract:** Total phosphorus (TP) concentration is high in countless small inland waterbodies in Hubei province, middle China, which is threating the water environment. However, there are almost no ground-based water quality monitoring points in small inland waterbodies, because the cost of time, labor, and money is high and it does not meet the needs of spatiotemporal dynamic monitoring. Remote sensing provides an effective tool for TP concentration monitoring spatiotemporally. However, monitoring the TP concentration of small inland waterbodies is challenging for satellite remote sensing due to the inadequate spatial resolution. Recently, unmanned aerial vehicles (UAV) have been applied to quantitatively retrieve the spatiotemporal distribution of TP concentration without the challenges of cloud cover and atmospheric effects. Although state-of-the-art algorithms to retrieve TP concentration have been improved, specific models are only used for specific water quality parameters or regions, and there are no robust and reliable TP retrieval models for small inland waterbodies at this time. To address this issue, six machine learning methods optimized by intelligent optimization algorithms (IOA-ML models) have been developed to quantitatively retrieve TP concentration combined with the reflectance of original bands and selected band combinations of UAV multispectral images. We evaluated the performances of models in terms of coefficient of determination ($R^2$), root mean squared error (*RMSE*), and residual prediction deviation (*RPD*). The results showed that the $R^2$ of the six IOA-ML models for training, validation, and test sets were 0.8856–0.984, 0.8054–0.8929, and 0.7462–0.9045, respectively, indicating the methods had high precision and transferability. The extreme gradient boosting optimized by genetic algorithm (GA-XGB) performed best, with the highest precision for the validation and test sets. The spatial distribution of TP concentration of each flight derived from different models had similar distribution characteristics. This paper provides a reference for promoting the intelligent and automatic level of water environment monitoring in small inland waterbodies.

**Keywords:** TP retrieval; IOA-ML models; UAV multispectral images; spatial distribution

## 1. Introduction

Inland waters are indispensable for agricultural, industrial, and recreational needs, such as aquaculture, transport, and energy production, and as a major source of drinking water and irrigation [1]. The deterioration of water quality has become one of the most important topics of environmental protection and the safe use of water. More than 60% of world's large lakes (>10 km$^2$) were considered eutrophic in the summer of 2012 [2]. According to the study of OECD (World Economic Cooperation and Development Organization), 80% of water eutrophication is attributable to phosphorus, and 10% is directly related to phosphorus and nitrogen [3]. In China, eutrophication of rivers and lakes has become more severe in the middle reaches of the Yangtze River, and phosphorus is the primary limiting element [4,5]. Therefore, monitoring the spatiotemporal variability of total phosphorus (TP) concentration is of great significance to protect the water environment.

Traditional water quality monitoring methods have great precision, mainly based on field sampling, laboratory analysis, or automated instruments [6]. However, these methods are labor-intensive, time-consuming, and costly, and do not meet the needs of spatiotemporal dynamic monitoring of water quality [1,7,8].

Over the last few decades, the role of remote sensing in water quality retrieval has been significantly increasing, with low-cost, full-coverage, and micro-dynamic characteristics thanks to the rapid growth in technologies and applications. For many years, satellites equipped with various sensors have been adopted for water quality assessment [9,10]. However, owing to the long return visit period, low spatial resolution, and susceptibility to interference by clouds, the application of satellites to real-time monitoring water quality of complex environments and small-sized waterbodies, such as small ditches and ponds, is not very suitable. Additionally, researchers have proved that small lakes were more vulnerable to eutrophication [11], and chlorophyll-a (Chl-a) concentrations were inversely related to lake size in the middle and lower reaches of Yangtze River [12].

Unmanned aerial vehicles (UAV) have led to innovative, regional monitoring of inland surface water and have successfully compensated for deficiencies in spatiotemporal resolution with flexibility and nonsusceptibility to interference by clouds [6,7]. UAVs equipped with multi-sensors, especially multispectral sensors, have been used to monitor Chl-a, total suspended solids (TSS), total nitrogen (TN), total phosphorus (TP), chemical oxygen demand (COD), and permanganate index ($COD_{Mn}$) in complex inland waterbodies [6,13]. For example, Su and Chou [14] used multispectral sensor mounted on a UAV to map the trophic state of a small reservoir. Wang et al. [15] designed an acquisition scheme of water quality spectral elements suitable for the complex waterbodies of aquaculture, combining the ground wireless sensor network and UAV spectral remote sensing technology. Liu et al. [16] constructed the inversion models of TP, TSS, and turbidity by multispectral sensor mounted on a UAV, and achieved higher accuracy through feature selection.

Although remote sensing can facilitate the monitoring of TP concentration, the methodology involved is complicated because phosphorus is nonactive and does not have spectral characteristics [17]. Therefore, the relationships between TP concentration and surface reflectance are nonlinear and complex [18]. The bands from visible to near-infrared have been used to estimate TP concentration [3,19]. The traditional retrieval models of water quality parameters based on statistical regression analysis, including linear regression, polynomial regression, ridge regression, and other methods, have poor inversion accuracy and weak generalization [20]. Machine learning (ML) methods are gaining momentum for water quality retrieval due to their ability to capture the potential relationship between remote sensing images and TP concentration [21,22]. In recent years, many researchers have proved that TP concentration can be estimated by ML methods with UAV multispectral images. For example, TP concentration in urban rivers was monitored by ML methods with UAV multispectral images [6]. Zhang et al. [23] developed a hybrid feedback deep factorization machine model to retrieve the concentration of phosphorus and trace pollution sources in urban rivers. Chang et al. [24] directly explored the TP spatiotemporal patterns with the aid of genetic programming models. UAV multispectral data were used to retrieve Chl-a, TN, and TP based on six ML models [25]. Based on the spectral and spatial features, Zhou et al. [26] used an ensemble ML model to estimate TP concentration in Shanghai. Although previous studies have used ML methods for TP estimation in different regions, specific models are only used for specific regions, or even only for the condition range in training data. In addition, almost all ML methods have their limitations, such as complex model hyperparameters. The intelligent optimization algorithm (IOA) can optimize the hyperparameters of ML methods due to their global search and adaptive characteristics and improve the robustness and predictability [27]. This paper establishes the TP retrieval models by combining the global search ability of IOA with the advantages of the high efficiency and flexibility of machine learning (IOA-ML) methods.

In this paper, six IOA-ML models were developed to retrieve TP concentration. By incorporating the UAV multispectral images, we attempt to propose methods for small inland waterbodies monitoring with high reliability and transferability. The main objectives of this study include (1) evaluating the performances of TP retrieval of six IOA-ML models with paired in situ data and UAV multispectral images divided into training, validation, and test sets, (2) conducting the statistical analysis of TP concentration based on pixel scale, (3) verifying the transportability of the developed IOA-ML models. This study is helpful for monitoring the water quality in small inland waterbodies and provides technical support for the intelligent management of water environment.

## 2. Materials and Methods

### 2.1. Study Area

In this paper, three typical small inland waterbodies on Hubei province, middle China, were selected as the research areas (Figure 1). Research area A and B are located in Jingmen city, Hubei province, with a linear distance of 12 km. Research area A is a drainage ditch for crayfish–rice culture, located in Zhanghe town, Dongbao district, and research area B is a small reservoir, located in Tuanlinpu town, Duodao district. The water quality of them both are mainly influenced by agriculture and aquaculture. Research area C is composed of six ponds, located in Shuangxiqiao town, Xian'an district, Xianning city. There is a domestic sewage collection and treatment facility, and the treated tailwater circulates among these ponds powered by engineering and is then discharged to the downstream after reaching the standard. The detail of research areas and sampling information are shown in Figure 1.

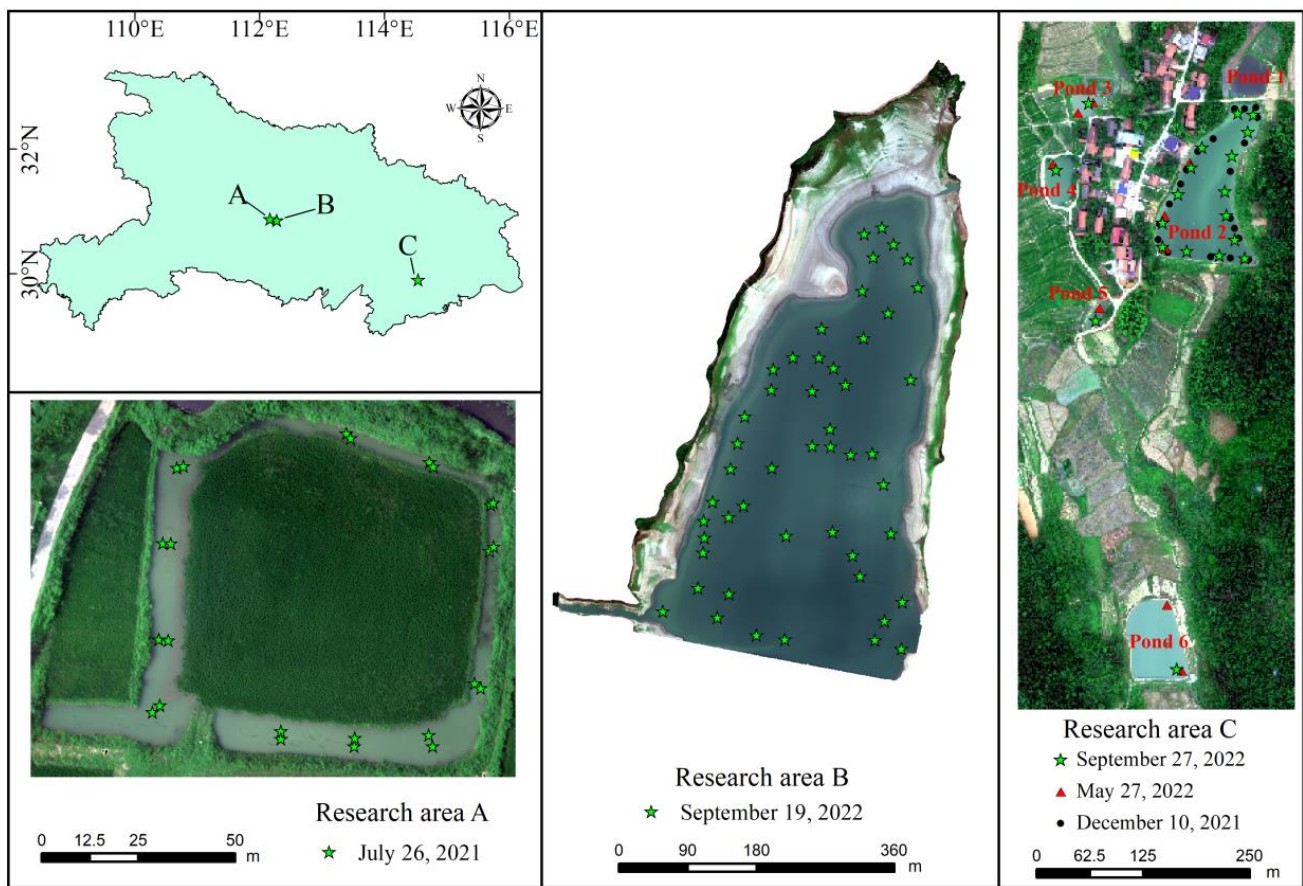

**Figure 1.** Locations and true color images from UAV multispectral images of the three research areas and geographical distribution of the sampling sites and times.

## 2.2. Data Processing

### 2.2.1. UAV Data and Preprocessing

The UAV platform utilized in our study is the DJI M300 RTK, manufactured by DJI innovations company, Shenzhen, Guangdong, China. The DJI M300 RTK is a 4-rotor UAV and integrates binocular vision, flight control system, and an FPV camera, with functions such as six-direction positioning, obstacle avoidance, and precise reshooting. It not only ensures flight safety but also provides necessary functions suitable for battery inspection applications [28]. The parameter details of this UAV are shown in Table 1. The multispectral imager mounted on the UAV is RedEdge MX Dual, manufactured by Micasense company, the United States. Ten multispectral bands can be obtained synchronously. In addition to the standard five-channel band of the RedEdge MX (first row of Table 2), a new five-channel sensor is added (second row of Table 2). Therefore, it is more suitable for water environment monitoring. The ground spatial resolution of multispectral images is 8 cm/pixel when the flight altitude of the UAV is 120 m. The ten-band range is shown in Table 2.

**Table 1.** Parameters of DJI M300 RTK.

| Item | Parameters |
|---|---|
| Diagonal wheelbases | 895 mm |
| Empty weight | 6.3 kg |
| Maximum takeoff weight | 9 kg |
| No load endurance | 55 min |
| Maximum flight/ascending/descending speed | 23 m/s/6 m/s/5 m/s |
| Maximum wind resistance level | 15 m/s |

**Table 2.** Band ranges of RedEdge MX Dual.

| RedEdge-MX Wavelength range (nm) | Blue475 $475 \pm 16$ | Green560 $560 \pm 13.5$ | Red668 $668 \pm 7$ | Red edge717 $717 \pm 6$ | Nir842 $842 \pm 28.5$ |
|---|---|---|---|---|---|
| RedEdge-MX Blue Wavelength range (nm) | Blue444 $444 \pm 14$ | Green531 $531 \pm 7$ | Red650 $650 \pm 8$ | Red edge705 $705 \pm 5$ | Red edge740 $740 \pm 9$ |

There were five flights in total from July 2021 to September 2022. The flight details are shown in Table 3. No ground control points were added to the flights because the multispectral imager has an integrated GPS that geo-tags each of the images acquired by the UAV. Furthermore, it was also equipped with a Downwelling Light Sensor and a Calibrated Reflectance Panel (CRP) to perform the radiometric calibration on the ambient light changes during the flight. A picture of the CPR was taken before and after each flight to capture the lighting conditions. The operational altitude of the UAV was 50–200 m, where atmospheric influence could be ignored. The multispectral images were mosaiced with radiometric correction after each flight. The water surface was extracted by the normalized difference water index (*NDWI*), calculated by Equation (1). The threshold was set to 0.2.

$$NDWI = \frac{Green560 - Nir842}{Green560 + Nir842} \tag{1}$$

where *Green*560 is the reflectance of the Green560 band and *Nir*842 is the reflectance of the Nir842 band.

**Table 3.** Flight and sampling information.

| Time | Area | Height | Resolution | Number of Photos | Sampling Number |
|---|---|---|---|---|---|
| 26 July 2021 | Research area A | 50 m | 3.67 cm | 1820 | 24 |
| 19 September 2022 | Research area B | 200 m | 14.4 cm | 3210 | 48 |
| 10 December 2021 | Research area C | 100 m | 7.7 cm | 6460 | 20 (all in pond 2) |
| 27 May 2022 | Research area C | 200 m | 14.3 cm | 2760 | 10 (4 in pond 2, 2 in pond 3, 1 in pond 4, 1 in pond 5, 2 in pond 6) |
| 27 September 2022 | Research area C | 200 m | 14.4 cm | 2560 | 19 (15 in pond 2, 1 in pond 3, 1 in pond 4, 1 in pond 5, 1 in pond 6) |

### 2.2.2. Field Data and Preprocessing

Water samples were collected in situ, as shown in Figure 1, synchronously with the collection of multispectral images in this study. Sampling points were 4–6 m away from the banks to avoid the influence of mixed pixels, except for research area C, which was evenly distributed on the water surface. Meanwhile, a real-time global positioning system was used to record the coordinate information of sampling points. According to the technical guidance for water quality sampling, the sampling points were arranged 0.1–0.2 m below the water surface because the water transparency was around 0.3 m, and 0.5 L of the water samples were collected at each point. Bottles were shaded before chemical analysis of TP concentration in the laboratory. After collecting water samples, the chemical experiment was completed within three days through a spectrophotometric analysis after the decomposition of potassium persulfate based on the Chinese national standard and trade standard. Two or three parallel samples of all water samples were analyzed, and the mean values served as the final TP concentration.

A single pixel can be easily impacted by specular reflection and water splash, making it difficult to reflect the spectral difference induced by an actual change in water quality at the sampling point [6]. For accurately matching UAV multispectral images with sampling points, a spatial window with $20 \times 20$ pixels was used to extract the reflectance of all bands located at each sampling point, rather than considering a single pixel in this study. The pixel values were extracted by the function of "Regions of Interest (ROI)" using EN-VI5.3.

### 2.3. Model Development

#### 2.3.1. Modeling Sets Construction

In this study, outliers deviating more than three standard deviations from the mean TP concentration were excluded, and 121 paired TP concentration and reflectance values were divided into training sets, validation sets, and test sets. The training sets and validation sets were used for model training, and test sets were not put into the model to measure the accuracy and generalization ability of the established models. In this paper, 20 sampling points were randomly selected as test sets, and the remaining 101 sampling points were divided into training and validation sets, with a ratio of 7:3. The model development process is shown in Figure 2.

#### 2.3.2. Feature Selection

To reduce the interference of background information and extract effective spectral information, it is important to try a variety of combined computing modes. Band ratio, a semiempirical method for retrieving water quality parameters, has been extensively researched and applied in monitoring inland waterbodies and has achieved promising results [29]. This method is often used in satellite remote sensing research to weaken the impact of atmospheric effects. The flight altitude of the UAV is sufficiently low that atmospheric effect could be ignored. Therefore, any 2–4 bands were combined as a feature through four fundamental admixture operations in this study, and 10,020 features were produced in total.

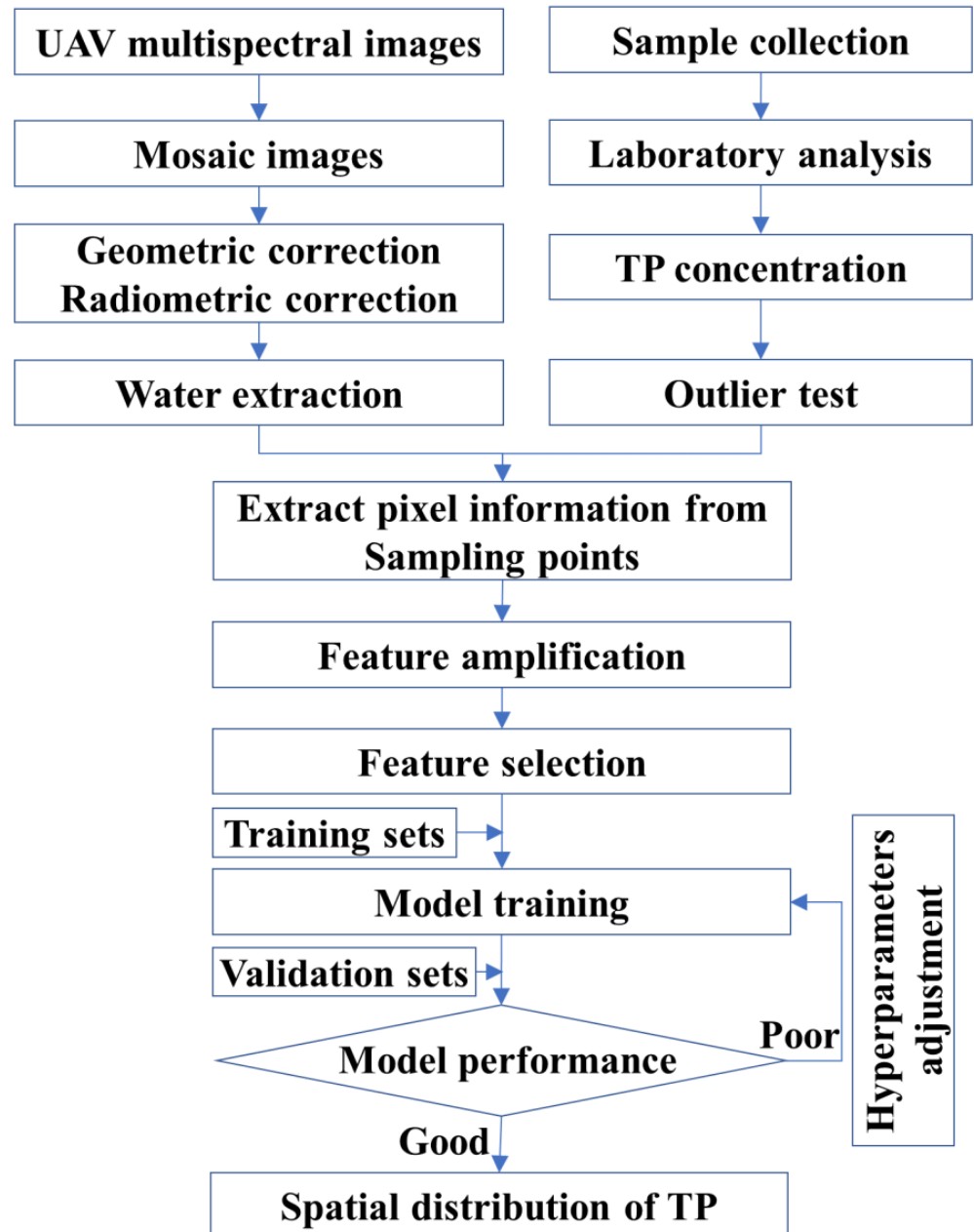

**Figure 2.** Flowchart of Model Development.

However, some features may be redundant because the information they add is contained in other features [30]. A good feature subset contains features that are highly related to target variable but are not related to each other. Feature selection is helpful to avoid overfitting and therefore improve model generalization. To better exploit the complex interaction between TP concentration and the reflectance of multispectral images, correlation-based feature selection (CFS) was used to select the feature subset used for the IOA-ML models [31]. The feature subset selected by CFS and the original ten bands are selected as the input variables of all IOA-ML models. The steps of CFS are as follows:

Step 1: find the feature that has the highest r (Pearson's correlation coefficient) value with the target variable and the feature is the first variable of the feature subset.

Step 2: select the feature that maximizes $Merit_s$ calculated by Equation (2) and add the selected feature to the feature subset.

Step 3: repeat step 2 until the value of $Merit_s$ does not increase.

$$Merit_s = \frac{k\overline{r_{cf}}}{\sqrt{k + k(k-1)\overline{r_{ff}}}} \tag{2}$$

where $Merit_s$ is the heuristic "merit" of the feature subset, $k$ is the feature number of the feature subset, $\overline{r_{cf}}$ is the average of the correlation between the target variable and the feature subset, and $\overline{r_{ff}}$ is the average intercorrelation of any two features in the feature subset.

The feature subset and TP concentration were processed by Z-score normalization to reduce the difference of feature ranges with the following equation before model establishment.

$$X_i' = \frac{X_i - \overline{X}}{X_{std}} \tag{3}$$

where $X_i'$ denotes the normalized result, $X_i$ denotes the reflectance values or TP concentration of the training, validation, and test sets. $\overline{X}$ denotes the mean of the training and validation sets and $X_{std}$ denotes the standard deviation of the training and validation sets.

### 2.3.3. IOA-ML Models

Hyperparameter is a kind of predetermined parameter before the learning process, and their values directly affect the performance of the ML models [31]. To improve the robustness and predictability of the ML models, IOAs were used to optimize the hyperparameters of the ML models. Six IOA-ML models, including support vector regression optimized by particle swarm optimization (PSO-SVR), categorical boosting regression optimized by genetic algorithm (GA-CBR), gradient boosting regression optimized by GA (GA-GBR), deep neural network (DNN), extreme gradient boosting optimized by GA (GA-XGB), and random forest optimized by grid search algorithm (GS-RF), have been developed. The details of the six IOA-ML models were as follows.

SVR can effectively deal with small sample and nonlinear problems and is a useful ML regression method. The radial basis function was chosen as the kernel of SVR in this paper. The hyperparameters of SVR, including penalty parameters C, gamma, and epsilon, were optimized by PSO [32].

The CBR model is a new gradient boosting decision tree algorithm that can handle categorical features well. This algorithm can deal with categorical features during training time instead of during preprocessing time and allows the use of whole datasets for training [33]. The hyperparameters of CBR, including number of iterations, learning rate, maximum tree depth, and regularization, were optimized by GA in this paper.

The GBR model is designed based on boosting. The GBR algorithm rebuilds the model in the gradient descent direction of the loss function of the previous iteration. Generally, the smaller the loss function, the better the model performance. The hyperparameters of GBR, including learning rate, number of estimators, subsamples, and maximum tree depth, were optimized by GA in this paper.

The XGB model, which was developed in 2016, is based on regression trees [34]. It improves the operational efficiency of the optimization process while reducing overfitting by employing second-order derivative data and integrating a regular component in the cost function. In this study, the hyperparameters of the XGB model, including learning rate, number of estimators, maximum tree depth, and minimum leaf weight, were optimized by GA.

DNN is the basic form of deep learning and one of the most efficient and powerful tools to model complex nonlinear relationships. DNN is a connectionist system with multiple hidden layers between the input and the output layers [35]. For tuning hyperparameters in this study, LeakyReLU and adam were set as the active function and optimizer, respectively. The neural units of each layer of the DNN model were (64, 128, 256, 512, 512, 1024, 1024). Additionally, dropout, batch normalization, and early stopping techniques were used in the model.

RF is an ensemble ML algorithm based on decision trees developed in 2001 by Leo Breiman [36]. RF is widely used in regression analysis, because of its high prediction accuracy. The GS algorithm tries the list of all combinations of values given for a list of hyperparameters and records the best performance based on evaluation metrics. The hyperparameters of RF, included number of estimators, maximum number of features, minimum number of samples to split a node, minimum number of samples to be at a leaf node, and maximum allowable depth, were optimized by GS.

### 2.3.4. Model Accuracy Assessment

To verify the performances of the models, we adopted three evaluation indicators: coefficient of determination ($R^2$), root mean square error (*RMSE*), and residual prediction deviation (*RPD*) to evaluate the accuracies of the TP retrieval models. Among them, $R^2$ is the most commonly used indicator to evaluate the performance of regression models. The value range of $R^2$ is [0, 1]. The *RMSE* indicates the relative error between the predicted value and the measured value. The closer it is to zero, the better the fitting model. The *RPD* is the ratio of standard deviation of measured value to MSE. The models can be divided into three categories according to *RPD*: (1) *RPD* > 2 indicates the model is stable and reliable; (2) 2 > *RPD* > 1.4 indicates the model is general and reliability needs to be improved; (3) 1.4 > *RPD* indicates poor stability of the model and retrieval is unreliable [37]. The formulas of these evaluation indicators are as follows, respectively.

$$R^2 = 1 - \frac{\sum_{i=1}^{n}(x_i - y_i)^2}{\sum_{i=1}^{n}(x_i - \overline{x})^2} \tag{4}$$

$$RMSE = \sqrt{\frac{1}{n}\sum_{i=1}^{n}(x_i - y_i)^2} \tag{5}$$

$$RPD = \frac{\sqrt{\frac{1}{n}\sum_{i=1}^{n}(x_i - \overline{x})^2}}{RMSE} \tag{6}$$

where $n$ is the number of data pairs, $y_i$ is the predicted TP concentration, $x_i$ is the value of the measured TP concentration, and $\overline{x}$ is the mean of measured TP concentration.

## 3. Results

### 3.1. Spectral Response to TP Concentration

The reflectance curves of sampling points are illustrated in Figure 3. The spectral signature was characterized by a strong absorption peak in blue444 and red668, and by a reflection peak in green560 and rededge705, which had similar characteristic with Lu et al. [29]. However, the TP concentration showed different laws with the reflectance values in research area A and B (Figure 3a) and research area C (Figure 3b). Overall, the reflectance curves were positively correlated with the TP concentration in research areas A and B, while the reflectance curves had a negative correlation with the TP concentration in research area C. This phenomenon can be explained by different water components or concentration in different regions, such as TP concentration and relative active parameters. It is worth noting that the water in research areas A and B is mainly affected by agriculture and aquaculture, while in research area C it is mainly affected by domestic sewage. The relationships between TP concentration and reflectance values in different regions were variable in previous studies. Research has demonstrated that there was a strong positive correlation between concentration of phosphorus and spectral reflectance [7,38]. Other studies showed that the spectral reflectance was inversely proportional to TP concentration in the lakes and rivers [25,39].

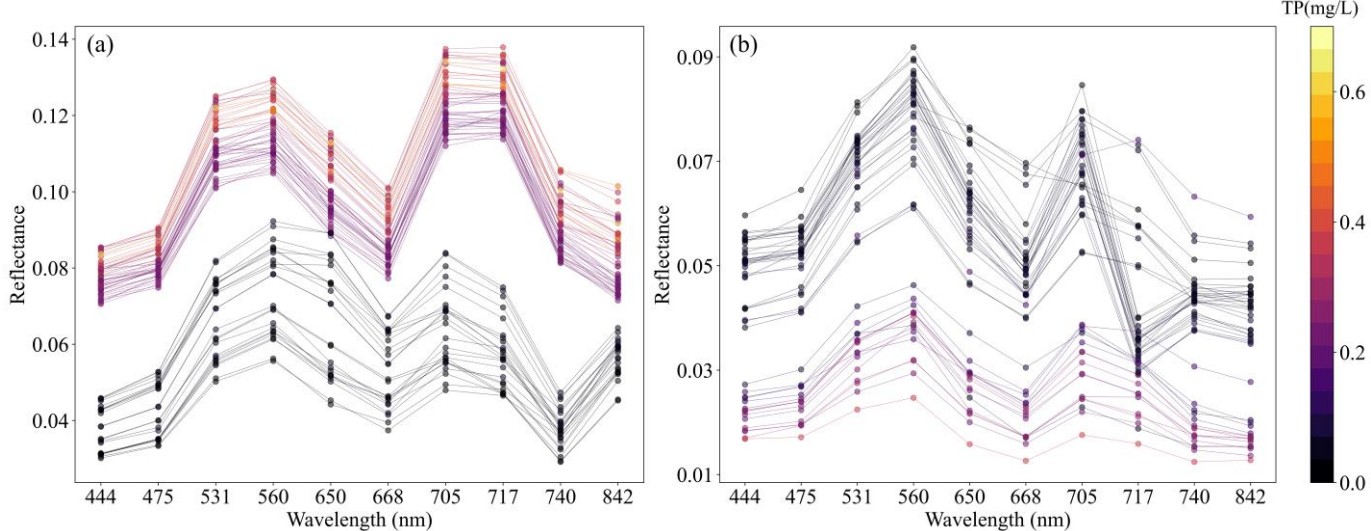

**Figure 3.** Reflectance curves of sampling points: (**a**) research areas A and B, (**b**) research area C.

### 3.2. Selection of Band Combinations

Band combinations were used to obtain sensitive bands to extract effective spectral information. The approach in Section 2.3.2 was used to create and select the best feature subset based on the training and validation sets for the input of the IOA-ML models. The best feature subset selected by CFS with maximum $Merit_s$ is shown in Table 4. The selected five-band combinations along with the ten initial multispectral bands were used as input variables in the development of the six IOA-ML models for retrieving TP concentration.

**Table 4.** Bands combinations selected by CFS.

| Selection Order | Feature | $Merit_s$ |
| :---: | :---: | :---: |
| 1 | rededge740 × rededge714 × rededge705 × nir842 | 0.8329 |
| 2 | (blue444 − red668 − nir842) ÷ rededge714 | 0.8796 |
| 3 | rededge740 × red668 × red650 × nir842 | 0.8952 |
| 4 | red668 ÷ green560 ÷ green531 ÷ rededge714 | 0.923 |
| 5 | rededge740 × rededge714 × red650 × nir842 | 0.9394 |

Note, × represents multiply, ÷ represents divide, − represents subtract.

### 3.3. Evaluation of IOA-ML Models

Six IOA-ML models (PSO-SVR, GA-CBR, GA-GBR, DNN, GA-XGB, GS-RF) with hyperparameters determined by the training sets and slightly adjusted by the evaluation indicators of the validation sets were developed to retrieve TP concentration. The scatter plots of measured and predicted TP concentrations on the training sets, validation sets, and test sets, derived from the developed IOA-ML models, are shown in Figure 4. The scatters were near and uniformly distributed on both sides of the 1:1 line, which proved the established models obtained better results. However, there still existed certain problems. For example, some models, such as PSO-SVR, GA-GBR, DNN, GA-XGB, and GS-RF, did slightly underestimate the high TP concentration where TP concentration was more than 0.7 mg/L due to the paucity of samples available with a high TP concentration.

Table 5 reveals the performances of six IOA-ML models with respect to $R^2$, *RMSE*, and *RPD*, and the best results are shown in boldface. In general, all models showed high accuracies in the training sets, validation sets, and test sets. The prediction accuracies for the validation and test sets were all lower than the prediction accuracies for their training sets for the GA-CBR, GA-GBR, DNN, GA-XGB, and GS-RF models. For the PSO-SVR model, the prediction accuracy of the training sets, validation sets, and test sets were similar. *RPD* are more than 2 of the six IOA-ML models and three sets, except for the test sets of the

GA-CBR model with 1.9849, proving the six IOA-ML models are stable and reliable on predicting TP concentration.

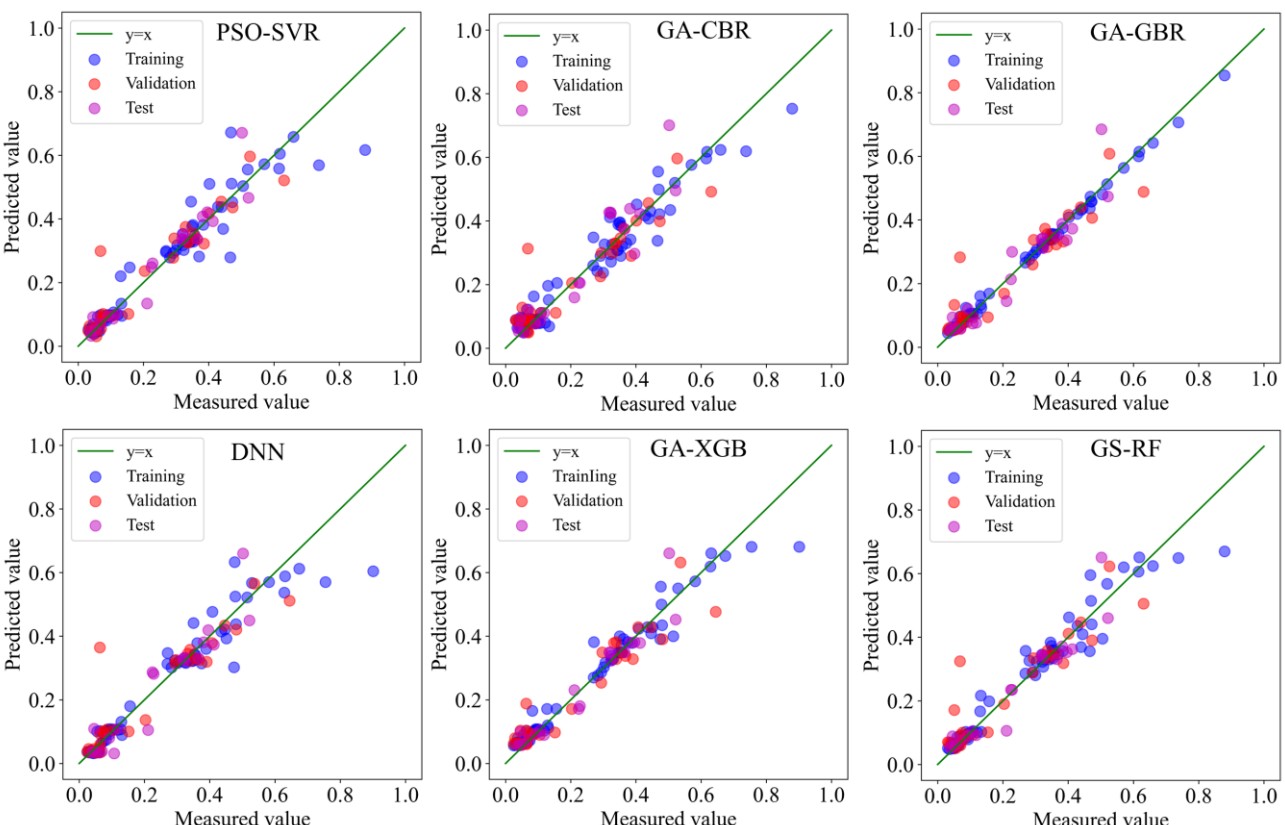

**Figure 4.** Comparison of measured and predicted TP concentrations of training sets, validation sets, and test sets.

**Table 5.** Performances of the six IOA-ML models.

| Model | $R^2$ | | | *RMSE* (mg/L) | | | *RPD* | | |
|---|---|---|---|---|---|---|---|---|---|
| | **Training** | **Validation** | **Test** | **Training** | **Validation** | **Test** | **Training** | **Validation** | **Test** |
| PSO-SVR | 0.9015 | 0.8929 | 0.9045 | 0.0649 | 0.0583 | 0.0486 | 3.1859 | 3.0561 | 3.2351 |
| GA-CBR | 0.9506 | 0.8148 | 0.7462 | 0.0445 | 0.0742 | 0.0792 | 4.5 | 2.3234 | 1.9849 |
| GA-GBR | **0.984** | 0.8458 | 0.8281 | **0.0253** | 0.0677 | 0.0651 | **7.9153** | 2.5466 | 2.4125 |
| DNN | 0.8856 | 0.8054 | 0.8143 | 0.0699 | 0.0786 | 0.0677 | 2.9565 | 2.2667 | 2.3206 |
| GA-XGB | 0.9584 | **0.9082** | **0.9124** | 0.0422 | **0.054** | **0.047** | 4.906 | **3.3005** | **3.379** |
| GS-RF | 0.9534 | 0.8579 | 0.8624 | 0.0447 | 0.0672 | 0.0583 | 4.6304 | 2.6528 | 2.6962 |

In this study, the paired data were divided into three sets, and the model performances on the validation and test sets were used to select the best retrieval model. Overall, the GA-XGB model outperformed the other models because it had the highest $R^2$ and *RPD* with the lowest *RMSE*. Many studies showed the XGB model had good performance on water quality retrieval, because the XGB model can control the model complexity and prevent the model from overfitting [1]. The accuracy of the water quality parameter retrieval model based on the GA-XGB algorithm also significantly higher compared with other methods [25]. As a second option, the PSO-SVR model also had better generalization capability and transferability, with the three evaluation indicators of the validation and test sets almost equal to the training sets, demonstrating the good fitting properties of the SVR model, even for non-linear data [18]. Additionally, Yang et al. [39] also concluded that the SVR model had the best performance for TP retrieval for Sentinel-2 in lakes and rivers. The

GA-CBR model presented the poorest performance on the validation and test sets, even with high $R^2$ for the training set compared to the other methods, which was overfitting in the training sets, to some extent.

### 3.4. Spatial Distribution of TP Concentration

The spatial distribution of TP concentration derived from the six established IOA-ML models with UAV multispectral images are mapped in Figures 5–7. Variability of TP concentration can be observed through the color change of each figure. The ranges of the color bars are retained consistently for better comparison for each flight. TP concentration derived from different IOA-ML models had similar spatial distribution characteristics. Statistical analysis of the retrieved TP concentration of each flight and the six IOA-ML models based on pixel scale and measured value are shown in Figure 8.

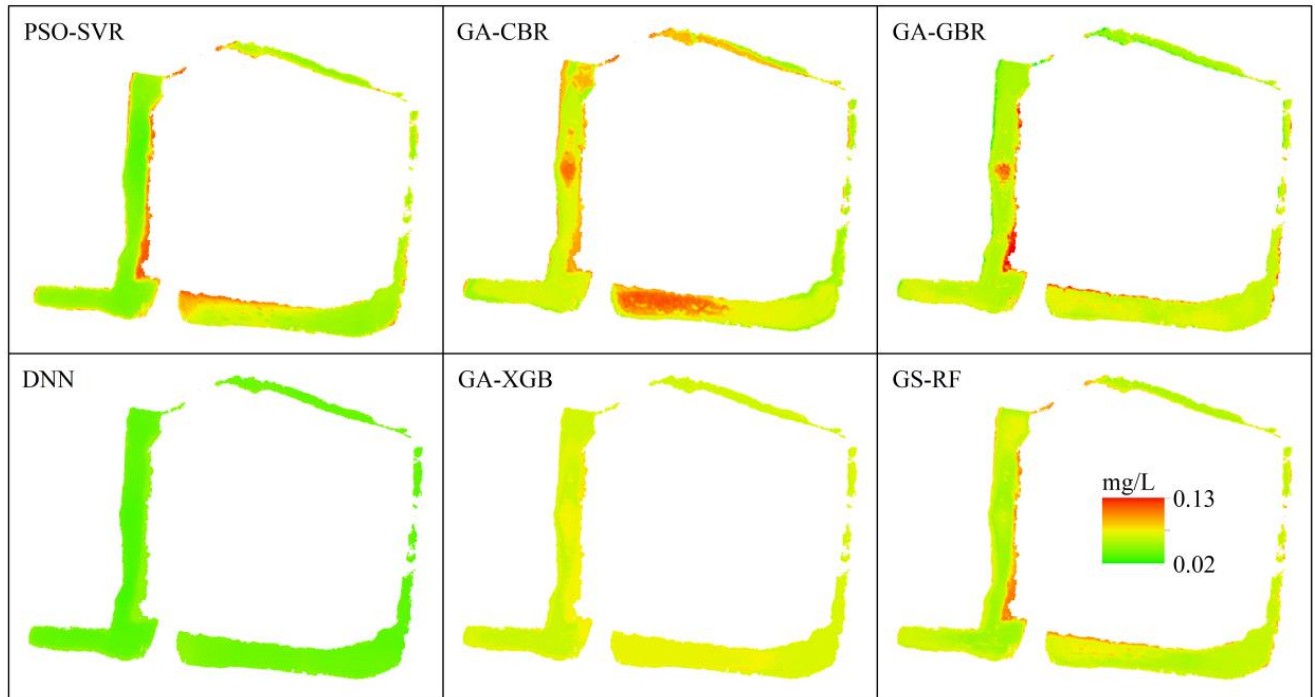

**Figure 5.** Spatial distribution of TP concentration derived from the six IOA-ML models in research area A.

The spatial distribution of TP concentration in research area A derived from the six IOA-ML models were consistent, to some extent (Figure 5). All retrieval models had high TP concentration in the center region of southern part and are more pronounced in the GA-CBR model. The measured TP concentration in the ditch of crayfish–rice culture was 0.046 ± 0.012 mg/L (mean ± standard deviation, hereinafter same). According to the statistics of the retrieval results of all pixels, the retrieved TP concentration of the PSO-SVR, GA-CBR, GA-GBR, DNN, GA-XGB, and GS-RF models were 0.0633 ± 0.0336 mg/L, 0.0752 ± 0.0195 mg/L, 0.067 ± 0.0251 mg/L, 0.0401 ± 0.0217 mg/L, 0.0693 ± 0.0079 mg/L, and 0.067 ± 0.0192 mg/L, respectively. Most of the retrieval models' predicted values were greater than the measured values, and only the performance of DNN was close to the measured value. The main reason for this was that the retrieval models overestimated the low TP concentration where the TP concentration lies in research area A.

The spatial distribution of TP concentration derived from the six IOA-ML models were highly consistent in most parts of research area B (Figure 6). The observed and retrieved TP concentration of the PSO-SVR, GA-CBR, GA-GBR, DNN, GA-XGB, and GS-RF models were 0.415 ± 0.113 mg/L, 0.3971 ± 0.128 mg/L, 0.3617 ± 0.1104 mg/L, 0.35 ± 0.1051 mg/L, 0.3908 ± 0.1165 mg/L, 0.3546 ± 0.097 mg/L, and 0.374 ± 0.0954 mg/L, respectively. It was obvious that the quantiles and mean value of the measured TP concentration were

slightly higher than all of the retrieved results in Figure 8b, with relative errors between predicted and measured mean values from −15.77% to −5.83%. This can be explained by the high TP concentration on the shore and low concentration in the center of research area B (Figure 6), and the number of sampling points near the shore were more than that of the center (Figure 1). However, TP concentration at the bank of the northern part derived from the six IOA-ML models showed huge difference, where the TP concentration of the PSO-SVR and DNN models was very high, while that of the GA-GBR and GA-XGB models was very low. We extracted the reflectance curves of this region and found that the reflectance curves are slightly more than the reflectance curves with high TP concentration in Figure 3a. Therefore, we thought the retrieval results at the bank of the northern part in research area B of the GA-GBR and GA-XGB models may be inaccurate.

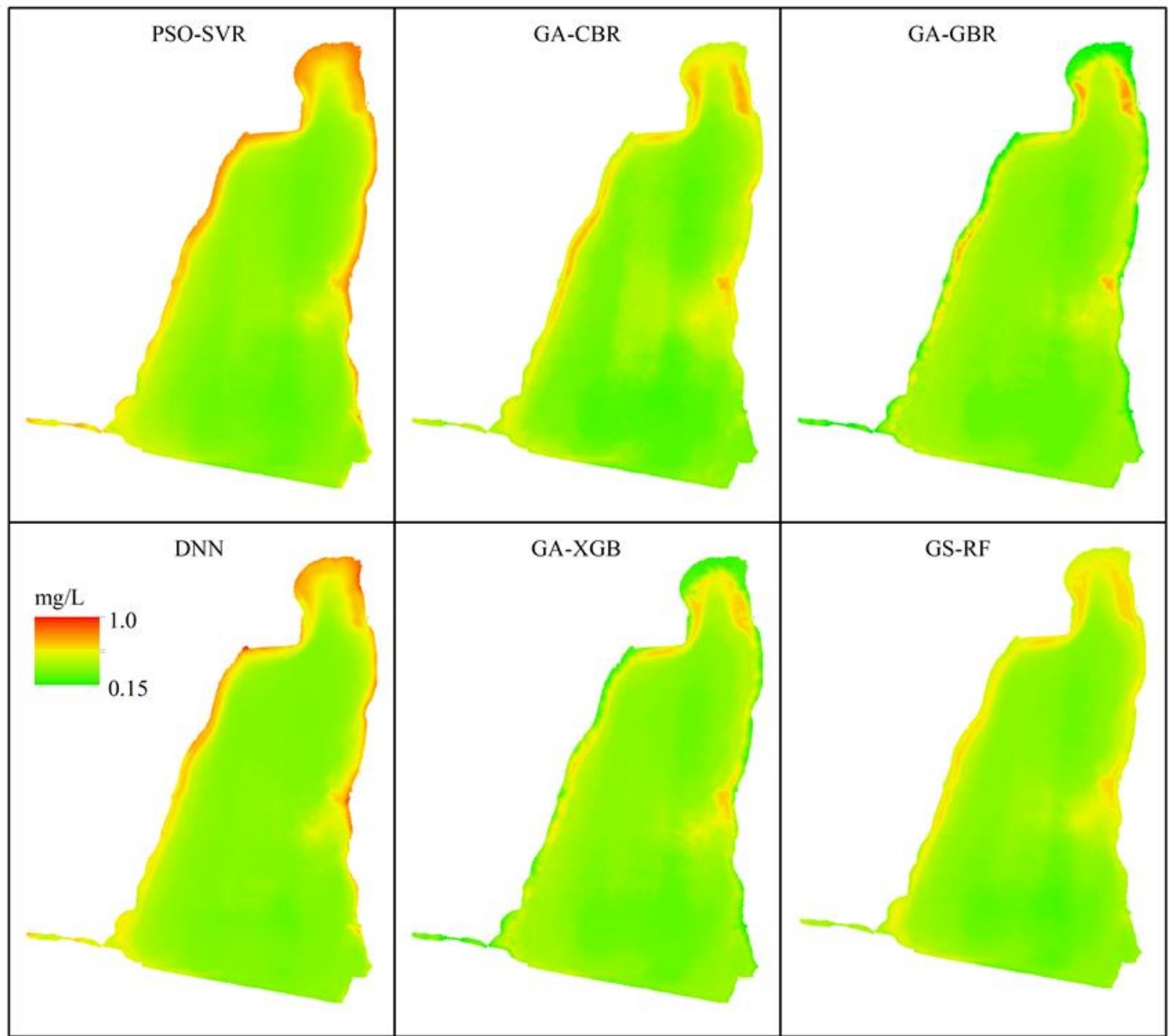

**Figure 6.** Spatial distribution of TP concentration derived from the six IOA-ML models in research area B.

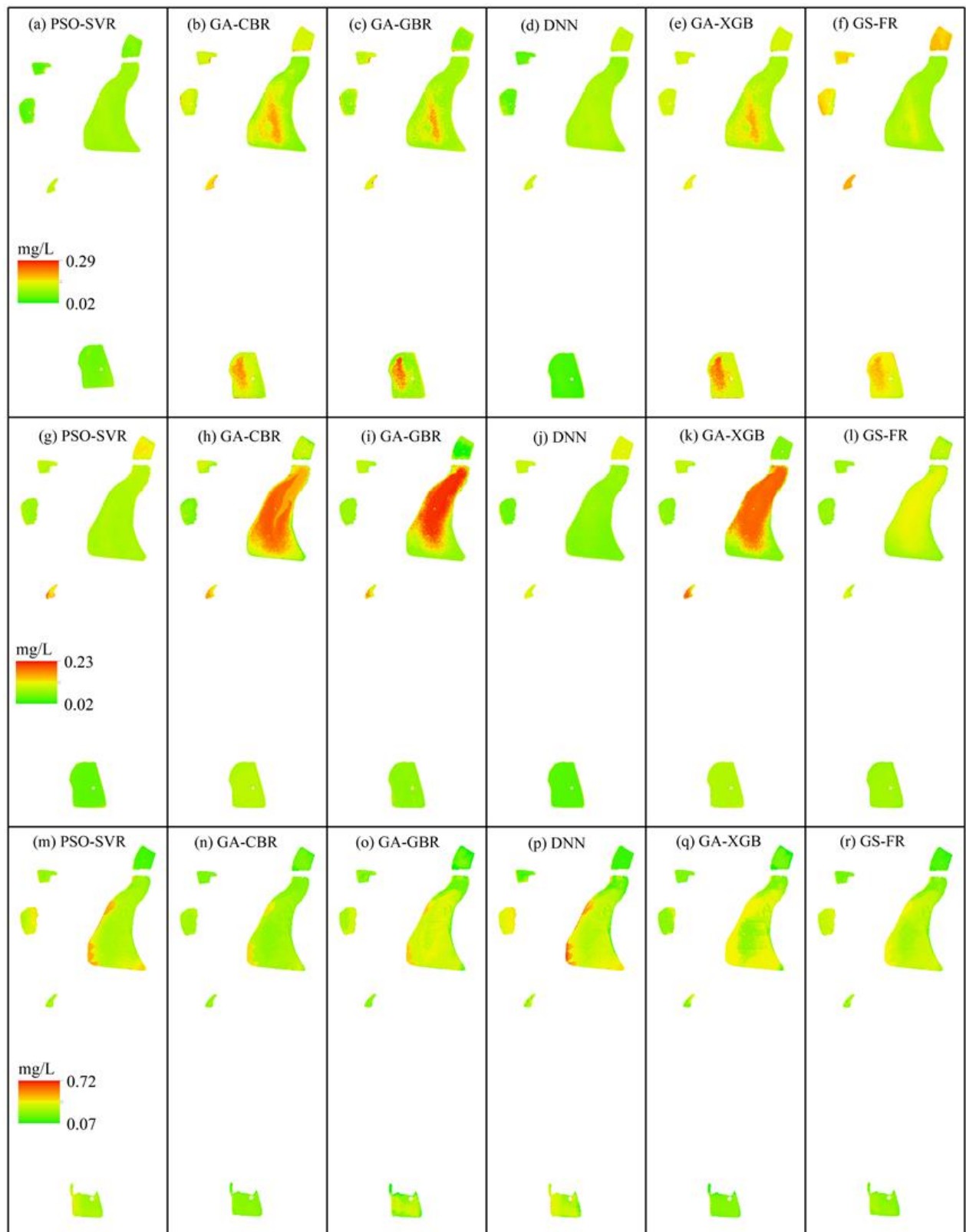

**Figure 7.** Spatial distribution of TP concentration derived from the six IOA-ML models in research area C on 10 December 2021 (first row (**a**–**f**)), 27 May 2022 (second row (**g**–**l**)), and 27 September 2022 (third row (**m**–**r**)), and every day has the same legends represented in (**a**,**g**,**m**), respectively.

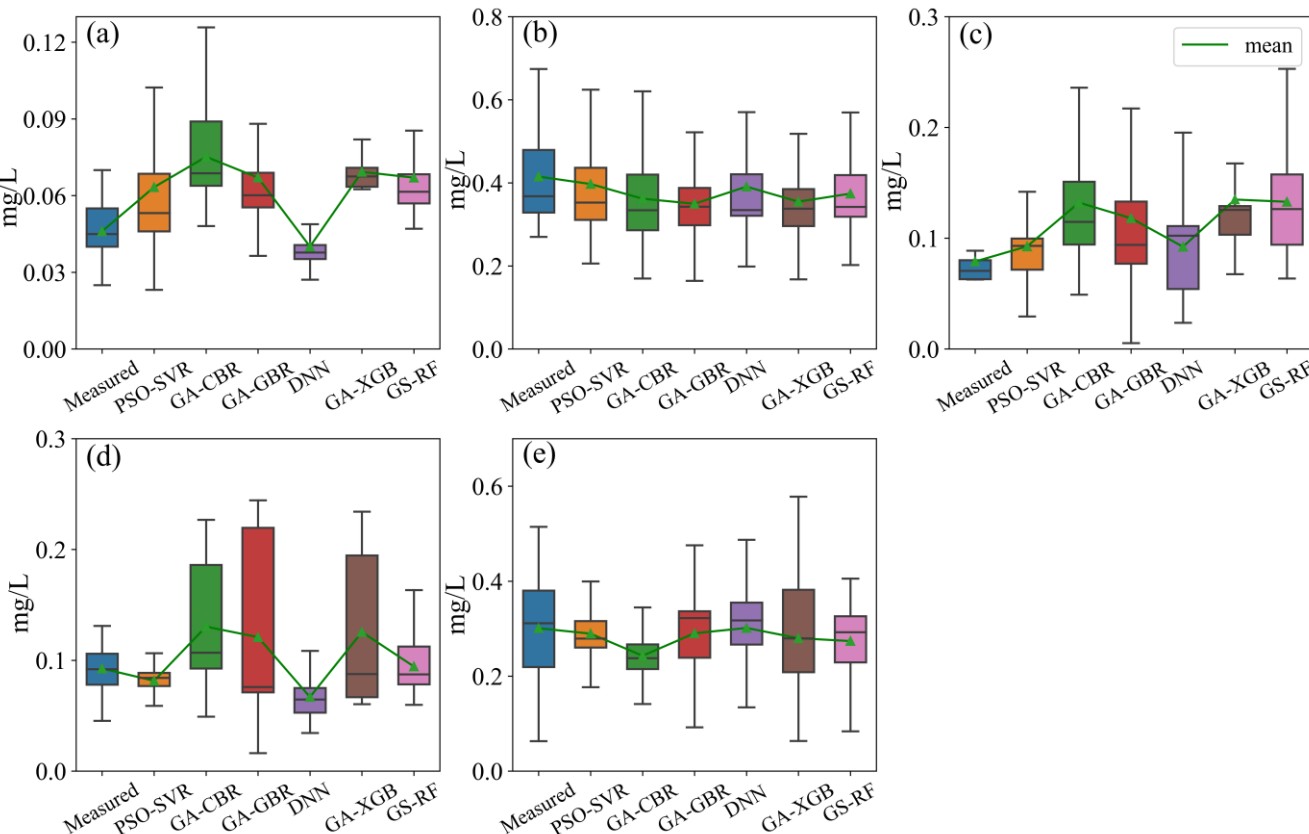

**Figure 8.** Statistics of measured TP concentration and derived TP concentration from the six IOA-ML models at the pixel scale: (**a**) research area A, (**b**) research area B, (**c**) research area C on 10 December 2021, (**d**) research area C on 27 May 2022, (**e**) research area C on 27 September 2022.

The TP maps derived from different IOA-ML models in research area C on 10 December 2021, 27 May 2022, and 27 September 2022 are presented in Figure 7. The visual inspection conveyed a high correspondence between different methods, though some differences were evident. The retrieval results delineated that the TP concentration in the center of Pond 2 and Pond 6 was high, while the near shore was low, and all the sampling points are near the shore of ponds on 10 December 2021. Therefore, the mean value of the measured values should be lower than the retrieval results, which was in accordance with Figure 8c. On 27 September 2022, all sampling points are located 4–6 m near the shore of ponds, with most in Pond 2. The results showed that the retrieved TP concentration at the shore of Pond 2 was high (Figure 7m–r). The measured values were high (Figure 8e) just right verifying the reliability of the retrieval results of the IOA-ML models. Therefore, sampling points should be evenly distributed throughout the study area in water quality parameter retrieval.

In research area C, there were a total of three flights. The TP concentration of the first and second flights were 0.02–0.28 mg/L and 0.02–0.23 mg/L, respectively, which were significantly lower than the third flight on 27 September 2022. The main reason for this was that the Yangtze River basin experienced continued drought in summer and autumn 2022, resulting in a great reduction of water volume in the ponds, and it was obvious that the water surface of Pond 6 on 27 September 2022 was almost half that of the previous. Studies proved that drought could increase water pollutant concentration. For example, researchers indicated that Chl-a concentration was negatively correlated with precipitation and water level in extreme drought events [40] and TP concentration in drought summer was significantly higher than those in the other years [41].

## 4. Discussion

Feature engineering before ML model establishment is necessary, and previous studies have confirmed that optimal input features can improve the performance of water quality retrieval models [12,42]. For example, there are generally perfect correlations between water quality parameters and band combinations of Landsat images [43]. However, there are only typically 4–7 multispectral bands in their study, and the feature selection is only based on the correlation between features and water quality parameters [6,10,16,42]. The multispectral imager used in this research has ten bands, and it is still unknow weather or not feature amplification and CFS can promote the model performance on TP retrieval in small inland waterbodies. The performances in the validation and test sets of the six IOA-ML models with feature amplification and CFS improved compared with those without feature amplification and CFS (Table 6). For validation sets, the *RMSE* decreased 5–18.91%, $R^2$ improved 0.49–6.62%, and *RPD* improved 1.5–16%. For the test sets, the *RMSE* decreased 1.04–30.11%, $R^2$ improved 0.2–7.29%, and *RPD* improved 0.95–23.15%. Among the six IOA-ML models, the DNN improved the most in the validation sets and the second most in the test sets, although the r values between the selected feature subsets and the measured TP concentration were 0.83, 0.66, 0.82, −0.12, and 0.83, respectively. The robustness of established IOA-ML models with feature amplification and CFS showed great improvement. Research indicated that CFS-PSO feature selection can identify and remove irrelevant variables [44] and revealed the superiority of the CFS procedure for the detection of optimal wavelengths [45]. Therefore, choosing a suitable feature subset by CFS can effectively improve the accuracy of the TP retrieval models.

**Table 6.** Improvement of the six IOA-ML models with feature amplification and CFS.

| Model | $R^2$ | | *RMSE* (mg/L) | | *RPD* | |
|---|---|---|---|---|---|---|
| | **Validation** | **Test** | **Validation** | **Test** | **Validation** | **Test** |
| PSO-SVR | 1.18% | 0.20% | −8.77% | −1.04% | 5.11% | 0.95% |
| GA-CBR | 5.61% | 7.29% | −16.30% | −14.12% | 14.07% | 12.32% |
| GA-GBR | 4.84% | 6.07% | −17.74% | −20.33% | 15.07% | 16.97% |
| DNN | 6.62% | 6.65% | −18.91% | −20.46% | 15.96% | 17.09% |
| GA-XGB | 2.53% | 6.65% | −11.85% | −30.11% | 10.57% | 23.15% |
| GS-RF | 0.49% | 4.72% | −5.00% | −20.45% | 1.50% | 16.93% |

One of the limitations of the ML-based TP retrieval model is that its transferability is limited [42]. Many studies used cross validation or splitting data into three sets to verify the feasibility of the ML-based water quality parameter retrieval models [17,25,26,46]. The paired TP concentration and reflectance values in this study were split into training, validation, and test sets, and model performances on the validation and test sets ($R^2$ = 0.7462–0.9124, *RMSE* = 0.047–0.0792 mg/L, *RPD* = 1.9849–3.379) showed a slight decline in different degrees compared to those on the training sets ($R^2$ = 8856–0.984, *RMSE* = 0.0253–0.0699 mg/L, *RPD* = 2.9565–7.9153), but the overall performances maintained a good balance. The results suggested that slight overfitting existed in the developed IOA-ML models, but it was controlled at a good level. Remotely-sensed TP estimation is complex, and Politi et al. [47] assessed 28 empirical algorithms sourced from the peer-reviewed literature using new satellite remote sensing data to identify the best water quality parameter retrieval algorithms in terms of accuracy and transferability and concluded that none of them exhibited satisfactory promise. One study showed that the best TSS retrieval model developed by a local dataset was accurate when applied to other areas [48]. Another UAV multispectral images without water sampling on 12 May 2022 in research area B were directly used to retrieve TP concentration combined with the established models. The retrieved TP concentration of the PSO-SVR, GA-CBR, GA-GBR, DNN, GA-XGB, and GS-RF models were 0.1273 ± 0.0346 mg/L, 0.1946 ± 0.0108 mg/L, 0.2356 ± 0.0117 mg/L, 0.151 ± 0.0273 mg/L, 0.224 ± 0.0226 mg/L,

and $0.1432 \pm 0.0111$ mg/L, respectively. The r values of TP concentration based on pixel scale derived from any two IOA-ML models were 0.4342–0.9083. Although the accuracy of the developed models may be mediocre when directly used in other flights under different external environments, such as temperature and sunlight intensity and hydrological regimes [22], it can still estimate the TP concentration of each pixel in the entire research area. In summary, the developed IOA-ML models have certain transferability at the research areas and can be easily applied to other regions by retraining the models with new data.

Although several approaches, including dividing the paired data into three sets and applying established models to other multispectral images, had been adopted to verify the transferability of the established models, the established models had not been verified using datasets in a separate waterbody. In addition, we only sampled at the edge of the ponds in research area C due to the constraints of financial support and time. For further research, we should collect more samples from various waterbodies and ensure that the sampling points are distributed as evenly as possible, and further establish more generalized and adaptable models.

### 5. Conclusions

In this study, six IOA-ML models were developed to retrieve TP concentration with UAV multispectral images in small inland waterbodies in Hubei province, middle China. The paired in situ TP concentration and reflectance values were divided into training sets, validation sets, and test sets. Feature selection was performed by CFS to find the most suitable feature subset. The developed IOA-ML models with hyperparameters tuned with training sets and slightly adjusted with validation sets achieved satisfactory performance in term of $R^2$ (0.7462–0.984), *RMSE* (0.0253–0.0792 mg/L), and *RPD* (1.9849–7.9153). The GA-XGB and PSO-SVR models had the best performances according to the accuracies of the validation and test sets. The TP concentration of each flight derived from six IOA-ML models had a similar spatial distribution, and quantiles and mean values of TP concentration based on pixel scale retrieved from the six IOA-ML models was lower than the measured value when most water sampling points were located in the high value area of the retrieval models. Additionally, the developed IOA-ML models have certain transferability at the research areas and can be easily applied to other regions by retraining the models with new data. This study provides an efficient and practical way for TP monitoring in small inland waterbodies.

**Author Contributions:** Conceptualization, W.H., W.G. and D.S.; methodology, W.H. and D.M.; software, W.H.; validation, W.G.; investigation, H.W., J.L. and H.W.; writing—original draft preparation, W.H.; writing—review and editing, W.G.; funding acquisition, D.S. All authors have read and agreed to the published version of the manuscript.

**Funding:** This study was supported by the Chinese National Natural Science Foundation (No. U21A20156) and the Strategic Priority Research Program of the Chinese Academy of Sciences (No. XDA2004030102).

**Data Availability Statement:** The raw data supporting the conclusions of this article are available from the authors upon request.

**Acknowledgments:** The authors would like to thank the reviewers and the editors for their valuable suggestions and contributions, which significantly helped to improve this article.

**Conflicts of Interest:** The authors declare no conflict of interest.

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
