# Peer review of "Retrieval of TP Concentration from UAV Multispectral Images Using IOA-ML Models in Small Inland Waterbodies"

_remotesensing, doi:10.3390/rs15051250_

Round 1

Reviewer 1 Report

The manuscript proposes a comparative analysis of six models to retrieve PT concentrations from a drone-borne multi-sensor camera for different ponds in China. The topic of the paper is interesting, but it is necessary to answer some missing points in the manuscript for more clarity and to make the work clearer to the reader. In general, the sections of the manuscript are clear and it can be considered for publication with some major corrections. My comments and suggestions are included in the pdf file.

Reviewer 2 Report

The paper presents the use of six different models to retrieve TP concentrations from ponds using UAV remote sensing. The work seems to be relevant for detecting pollution in small water bodies. However, I think that the manuscript needs a major revision. Please see attached.

Round 2

Reviewer 1 Report

To authors

Author Response

We greatly appreciate your professional evaluations and comments, which help to improve the quality of the manuscript greatly. This section and related content were removed after careful consideration in the revised manuscript. The revised content is marked up using the “Track Changes” function in the revised manuscript. Thank you again for your valuable suggestion.

Reviewer 2 Report

The authors largely agree with my comments and suggestions. It is now left to the editor to ensure that those suggestions reflect in the revised manuscript.

Author Response

We greatly appreciate your professional evaluations and comments, which help to improve the quality of the manuscript greatly. Thank you very much for your approval of this manuscript.